# TOWARDS CONTROLLABLE POLICY THROUGH GOAL-MASKED TRANSFORMERS

## ABSTRACT

Offline goal-conditioned supervised learning (GCSL) can learn to achieve various goals from purely offline datasets without reward information, enhancing control over the policy. However, we argue that learning a composite policy switchable among different goals seamlessly should be an essential task for obtaining a controllable policy. This feature should be learnable if the dataset contains enough data about such switches. Unfortunately, most existing datasets either partially or entirely lack such switching demonstrations. Current GCSL approaches that use hindsight information concentrate primarily on reachability at the state or return level. They might not work as expected when the goal is changed within an episode. To this end, we present Goal-Masked Transformers (GMT), an efficient GCSL algorithm based on transformers with goal masking. GMT makes use of trajectory-level hindsight information, which is automatically gathered and can be adjusted for various statistics of interest. Due to the autoregressive nature of GMT, we can change the goal and control the policy at any time. We empirically evaluate GMT on MuJoCo continuous control benchmarks and Atari discrete control games with image states to compare GMT against baselines. We illustrate that GMT can infer the missing switching processes from the given dataset and thus switch smoothly among different goals. As a result, GMT demonstrates its ability to control policy and succeeds on all the tasks with low variance, while existing GCSL works can hardly succeed in goal-switching[1].

## 1 INTRODUCTION

Runners can control and adjust their pace in a marathon by switching comfortably between various poses for different goals. Similarly, agents can also acquire such switching ability through reinforcement learning (RL) or imitation learning (IL). This process generally requires environments that can start with arbitrary pose states, carefully tuned rewards, or massive offline demonstrations. However, these critical things are notoriously challenging to obtain. In comparison, by knowing the pace of each running stance, a human can easily switch between various poses to control speed without learning such switching processes intentionally.

From another perspective, we try to formulate this problem as goal-conditioned supervised learning (GCSL) (Ghosh et al., 2019) problem given a fixed amount of offline dataset: Considering pose or pace as a goal, can agents learn a composite policy that can switch among these goals interchangeably over the dataset? We refer to this problem as the *goal-switching problem*. Since the distribution of initial states shifts between the training and evaluation, this problem might face the *covariate shift* issue. Instead of a fixed set of states in the training set, any state might be the start of the switched goal during evaluation, resulting in agents not knowing how to achieve the goal. The goal-switching has widespread adoption in practical applications. The control of robots to transfer to a different skill while performing another skill is essential in robotics. In the game field, it can induce immersive experiences by managing the performance and strategies of AI bots according to the game progress.

Recent works (Ghosh et al., 2019; Ding et al., 2019; Furuta et al., 2021; Eysenbach et al., 2022; Reed et al., 2022) on GCSL mainly focus on learning how to achieve arbitrary goals. They tend to use either state (Emmons et al., 2021) or return-to-go (Chen et al., 2021) as goals. Models that use states

---

[1]The code will be available as soon as possible.

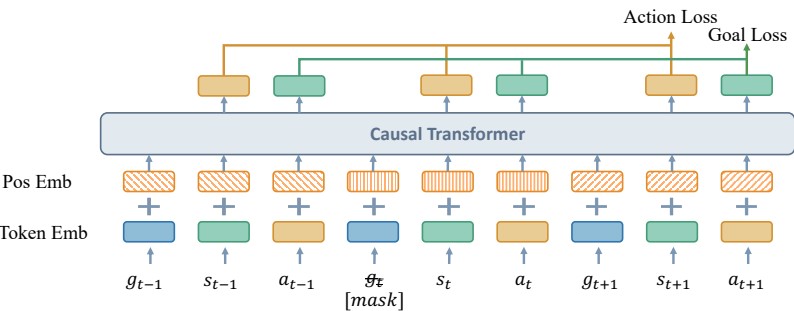

Figure 1: Overview of GMT architecture.

as goals can be called state-conditioned models. These models generate actions based on targeting future states or slices of successful demonstration. There are also models conditioning on return-to-go — the total rewards an agent can receive from the current step until the end of an episode. These methods typically use the "relabeling" strategy to excavate a variety of goal signals by bootstrapping any of the aforementioned goals from a fixed dataset. However, each method from above has some fatal flaws that make them result in poor performance on the goal-switching problem. By setting return-to-go as goals, switching can only happen by tweaking a normalized return-to-go, which is neither intuitive nor efficient. State-conditioned methods can reach an arbitrary state in theory. However, they tend to model the problem as a pure MDP, which leads to poor performance or failure when demonstrations of transiting between two states are missing from the dataset. We argue that the essence of the goal-switching problem is to enable the model to learn unseen transitions. Model-based approaches may be one remedy, which allows for unseen transitions by planning across a latent space (Jiang et al., 2022) or an explicitly learned world model (Micheli et al., 2022). These methods require additional environmental model learning with higher learning complexity.

This paper presents Goal-Masked Transformers (GMT), an efficient GCSL algorithm that neither necessitates explicit world model learning nor successful demonstrations to solve the goal-switching problem. In particular, we employ a causal transformer (Radford et al., 2019) to autoregressively describe trajectories with transitions consisting of goals, states, and actions. With such a setting, the goal can be changed at any moment, releasing the full potential of the policy control. However, since the goals are the same in each transition during training, the model tends to neglect the change of goals, leading to a goal-switching failure during evaluation. In order to compel the model to put more effort into goals, we introduce a masking mechanism with a probability of replacing the goal information with a $[mask]$ token. As a result, we observe the promising outcome that agents can smoothly switch from one goal to another. Figure 1 presents an overview of GMT.

Similar to previous GCSL works, we apply the "relabeling" strategy to increase the diversity and coverage of goals. Additionally, there are various expressions of goals and numerous approaches to achieving them. Thus, we propose a simple yet effective approach that automatically aggregates and clusters the offline data into several goals in accordance with the statistics of interests.

In summary, our main contributions are as follows:

- We draw attention to the goal-switching problem and argue that doing so is an essential step toward controllable policy. The goal-switching problem requires that agents should adapt to goal changes within one episode. It is an exceedingly challenging generalization problem, especially when training on limited datasets.

- We propose Goal-Masked Transformers (GMT), a family of goal-conditioned algorithms based on causal transformers with a goal-masking mechanism and hindsight information to achieve controllable policy. Through experiments, we demonstrate that GMT possesses goal-switching capabilities that are not present in the current GCSL algorithms.

- We introduce an unsupervised approach to cluster trajectories into multiple goals from the datasets without any goal information. Empirically, we find that this approach improves the stability and efficiency of the switching process.

## 2 METHOD

The purpose of Goal-Masked Transformers (GMT) is to train a controllable policy that can comfortably switch among goals over the demonstration $\{(s_0^i, a_0^i, s_1^i, \dots)\}_{i=0}^N$ where $s_t^i, a_t^i$ indicate state and action at timestep $t$ in the $i$th of $N$ trajectories respectively. To enable goal-switching, we design trajectories $\tau_i$ with transitions made up of goal, state, and action, i.e.,

$$\tau_i = (g_0^i, s_0^i, a_0^i, g_1^i, s_1^i, a_1^i, ..., g_{T_i-1}^i, s_{T_i-1}^i, a_{T_i-1}^i) \tag{1}$$

where $g_t^i$ denotes goal at timestep $t$ in the $i$-th trajectory and $T_i$ indicates the length of the trajectory $\tau_i$. We define the goal as statistics of interests, for example, reward, speed, number of blocks that break in a game, the dynamics of robots and strength of game bots. These statistics are generally not provided in the demonstration but are implicitly included in the data. The statistics can be linked to each trajectory by some kind of mapping function. Namely, we can mathematically formulate the goal as follows.

$$g = f(s_0^i, a_0^i, s_1^i, a_1^i, ..., s_{T_i-1}^i, a_{T_i-1}^i; e) \tag{2}$$

where $e$ indicates the statistics of an interest and $g$ indicates the goal. Given the statistics of a specific interest $e$, the function $f$ can map the trajectory into the corresponding goal.

### 2.1 TRAINING DATASET

In order to demonstrate the ability of GMT in different areas, we consider two applications to evaluate GMT. We train our model on both tasks by utilizing offline trajectory data collected from RL agents, either are trajectories rollouts during training or collected by using training checkpoints with a certain level of skills to inference.

**D4RL**   We conduct 4 MuJoCo (Todorov et al., 2012) locomotion environments with continuous control from the D4RL (Fu et al., 2020) benchmark, including HalfCheetah, Hopper, Walker, and Ant. The locomotion task is the fundamental task in the field of robotics and we can show the control of the movement speed (along the x-axis) by switching target poses. We combine the Medium and Medium-Expert datasets provided in the D4RL benchmark as the training dataset for each locomotion environment except for HalfCheetah. On the HalfCheetah task, we also add the Random dataset where trajectories are totally generated by a random policy.

**Atari**   We demonstrate that GMT can also be applied to games, such as video games. The RLUnplugged (Gulcehre et al., 2020) dataset provides trajectories of numerous Atari games (Bellemare et al., 2013) collected from the training progress of a DQN (Mnih et al., 2013) agent. The dataset covers various demonstrations, including agent behavior at all learning stages. We train GMT over the Breakout dataset to show that GMT can switch between different levels of policies according to the changes in goal. The method should be able to apply to any game in theory, but due to the computational limitation, we only randomly select one game to show the idea.

The trajectories in both the public datasets contain states, actions, and rewards. However, we focus on goals instead of directly utilizing the reward information. Commonly in practice, the reward information is hard to obtain in sparse rewards settings or requires sophisticated designs of reward functions. Therefore, we leave reward information and use the "relabeling" strategy to extract goal information from the trajectories based on any statistics of interest to construct new trajectories with transitions consisting of goals, states, and actions.

A straightforward strategy to relabel the trajectory with goals is to manually group statistics of interest by hand. However, we observe that too few classes may lead to a high variance policy as extremely dissimilar demonstrations are grouped. Whereas having too many classes can hinder performance because the model struggles to distinguish similar trajectories. There might eventually be relatively few demonstrations that are rarely learnable. Thus, we require a novel relabeling approach that can automate this process in an efficient and effective way.

Here we propose an unsupervised manner to aggregate and label trajectories for the aforementioned offline datasets. We first perform a k-means clustering (Lloyd, 1982) along the statistics of interest (e.g. we are interested in the speed of locomotion on the x-axis in MuJoCo) with a range of $k$, and then calculate the silhouette score (Rousseeuw, 1987) of each cluster. As there are now goodness

measurements of clustering with different $k$, we sample a $k$ based on the normalized silhouette score from the top 85th percentile.

We uniformly sample 2000 trajectories to train models for each Atari game. For MuJoCo tasks, the specific numbers of trajectories in the dataset are listed in Appendix A.2.1. Moreover, we do not require any extra dataset balancing or filtering since the major purpose of this research is to show that our model can switch between multiple goals rather than the magnitude of the rewards that agents will ultimately receive.

## 2.2 ARCHITECTURE

GMT accepts input in the form of a transition consisting of a goal, state, and action token. The tokenization scheme used for GMT is described in Appendix A.1.1. To generate the final model input, we apply three parameterized embedding functions $f(\cdot; \theta_g^e), f(\cdot; \theta_s^e), f(\cdot; \theta_a^e)$. The embedding functions carry out various operations depending on the modality of the token to enable efficient learning from the multi-modal input sequence. A learned vector embedding space is used to store tokens associated with discrete- or continuous-valued observations or actions for any timestep. A single Convolutional Neural Network (CNN) is used to embed tokens from images at any timestep. Learnable position encodings are added for all tokens based on their global token position within their corresponding trajectories.

Under these circumstances, we try to formulate this problem as a sequence modeling problem. Due to the nature of the given sequence, we use an ordinary causal transformer or decoder-only transformer (Vaswani et al., 2017) as our main architecture. We feed $3K$ tokens worth of the most recent $K$ timesteps (goal, state, action for each timestep) into GMT, where $K$ represents the context length of GMT. In general, a longer context length allows the model to trace back longer in history but may require a more competent model and can be harder to train. A LayerNorm (Ba et al., 2016) is added before feeding the tokens to GMT.

**Loss** Given goals $g_{\leq t}$, states $s_{\leq t}$ and actions $a_{<t}$, GMT predicts the action $a_t$ as $\hat{a}_t$ and calculate the difference between $\hat{a}_t$ and $a_t$. For discrete (goals and actions for Atari games) and continuous values (actions for MuJoCo tasks), GMT respectively uses cross entropy and mean square error (MSE) as the loss. When only the supervisory signal of the action is there, the relationship between goals, states, and actions is frequently neglected. This situation is particularly evident in the case of goals because goals share the same value along the trajectory. In order to improve the situation, we add an auxiliary goal loss to force the model to focus on goal information. Therefore, the total loss $\mathcal{L}$ is a weighted sum of the action loss $\mathcal{L}_a$ and goal loss $\mathcal{L}_g$, namely, $\mathcal{L} = \mathcal{L}_a + \alpha \mathcal{L}_g$ where $\alpha$ is the scalar weight to balance two terms. We set $\alpha = 1$ at our all experiments. It is worth noting that there is no loss for the first goal as it is always given to the model.

## 2.3 GOAL MASKING

Surprisingly, the naive goal prediction task hardly contributes to policy learning. The model only requires the goal information to predict the goal label at the early stage of the trajectory. When reaching several goals, the model can actually predict the goal label from the states and actions as the state information may differ significantly across different goals. The model tends to ignore the goal label in the later phase of the training data and only learns the relationship between states and actions. The goal prediction task, therefore, becomes a goal memorization task.

Thus, instead of always using the actual label when predicting the goal, we have a probability of $p$ to replace the true goal with a trainable token, $[mask]$. We expect the model to learn and enforce the relationship between goals, states, and actions instead of only memorizing the previous goals. We propose *token masking* that replaces each goal token with a $[mask]$ token with a probability of $p$, but only performs within a context length[2]. All the masking operations are dynamic, meaning the mask is re-sampled for each training iteration. We conduct experiments to compare different $p$ (See

---

[2]We also propose *trajectory masking* that replaces goal tokens in a single trajectory $\tau$ with a probability of $p$. However, we find that this method still struggles with the goal-switching problem and cannot obtain a controllable policy. We analyze the reasons in Section 2.3

Table 1: Success rate of GMT and other baselines on different MuJoCo task

|  | HalfCheetah | | | Hopper | | Walker | | Ant | |
| --- | --- | --- | --- | --- | --- | --- | --- | --- | --- |
|  | $1 \to 0$ | $2 \to 0$ | $2 \to 1$ | $0 \to 1$ | $1 \to 0$ | $0 \to 1$ | $1 \to 0$ | $0 \to 1$ | $1 \to 0$ |
| GMT(token masking) | **15/15** | **15/15** | **15/15** | **15/15** | **15/15** | **14/15** | **14/15** | **14/15** | **13/15** |
| GMT(traj. masking) | 3/15 | 0/15 | 0/15 | **15/15** | 1/15 | **14/15** | 3/15 | **14/15** | 2/15 |
| GMT(no masking) | 0/15 | 0/15 | 0/15 | **15/15** | 0/15 | **14/15** | 0/15 | **14/15** | 0/15 |
| RvS(Emmons et al., 2021) | 0/15 | **15/15** | 7/15 | 0/15 | 0/15 | 3/15 | **14/15** | 2/15 | 1/15 |
| Prompt DT (Xu et al., 2022) | 6/15 | 2/15 | 4/15 | 1/15 | 3/15 | 1/15 | 0/15 | 2/15 | 3/15 |
| Decision Transformer (Chen et al., 2021) | 5/15 | **14/15** | 3/15 | 9/15 | 5/15 | 8/15 | 4/15 | 1/15 | 1/15 |

Section 3.2.1) and eventually find that token masking with $p = 0.5$ works the best. We set it as the default for our method.

# 3 EXPERIMENTS

We use a unified architecture for both Atari games and MuJoCo tasks (See Appendix A.1.2 for the implementation details and A.1.3 for deployments). In this section, we mainly demonstrate the results of MuJoCo tasks and analyze how GMT works through a series of ablation studies. We leave the analysis of Atari games in the Appendix (see A.2.2).

## 3.1 RESULTS

In this section, we use both step-based graphs and task success rates to evaluate the ability of GMT. Notice that higher variance may not necessarily indicate destructive behaviors in step-based illustrations. We focus on trajectory-level alignments, meaning different trajectories with the same goal can eventually reach similar final results through different paths. Thus, all the following step-based plots are specifically used to demonstrate consistent differences between various goals. The success rate is summarized as tables for each task.

We also summarize the success rate of each MuJoCo task for different models in 1. We define each task as a success if the model can successfully achieve the first goal and consequently switch to a new goal, where the difference in statistics of interests is within $10\%$.

### 3.1.1 MUJOCO

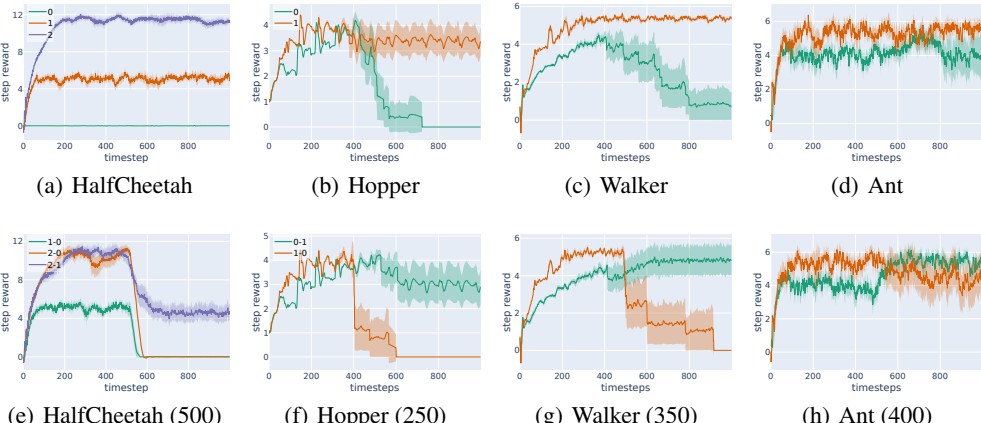

(a) HalfCheetah    (b) Hopper    (c) Walker    (d) Ant

(e) HalfCheetah (500)    (f) Hopper (250)    (g) Walker (350)    (h) Ant (400)

Figure 2: GMT with non-switching (a-d) and switching (e-h) tasks. Numbers in brackets in the switching task represent the timestep that goal switching happened. The number of lines represents the number of goals for each task, which is obtained by our automatic clustering method. We also experiment with the effect of generating more goals, see A.2.7 for detailed results.

We use four MuJoCo environments (HalfCheetah, Hopper, Walker, and Ant) as our major testbed to evaluate GMT. In all environments, our statistic of interest is the speed on the x-axis (which is also the reward of the environments at each step on the default setting).

All figures in this section (unless otherwise mentioned) use concrete lines to represent the mean and the same color with higher transparency around lines to indicate the standard deviation of the statistics. All the mean and variance are obtained by running over the model 15 times with 15 fixed random seeds. We show that GMT is capable of learning to imitate different trajectories with sufficient data in Table 3. We also demonstrate step rewards received across different goal labels and tasks in Figure 2.

As depicted in Figure 3, we use t-SNE (Hinton & Roweis, 2002) to visualize the distribution of states for trajectories with different goals. We only demonstrate the results of HalfCheetah here, full results including other environments can be interpreted similarly and can be found in Appendix A.2.4. From the figure, some states' overlap can be observed as many of these states may start from similar points or cross similar middle states when reaching desired goals. To mathematically model the problem, the naive goal prediction task becomes to model the probability of $P_\theta(g_t|s_\kappa, a_\kappa, g_\kappa)$, where we use the footnote $\kappa$ to represent indices in the context of our model. However, when states are distinct enough, which is very common in the later simulation of the MuJoCo task, the goal prediction task tends to degenerate to $P_\theta(g_t|s_\kappa, a_\kappa)$ as previous goals are no longer necessary. This also explains why trajectory masking does not work well.

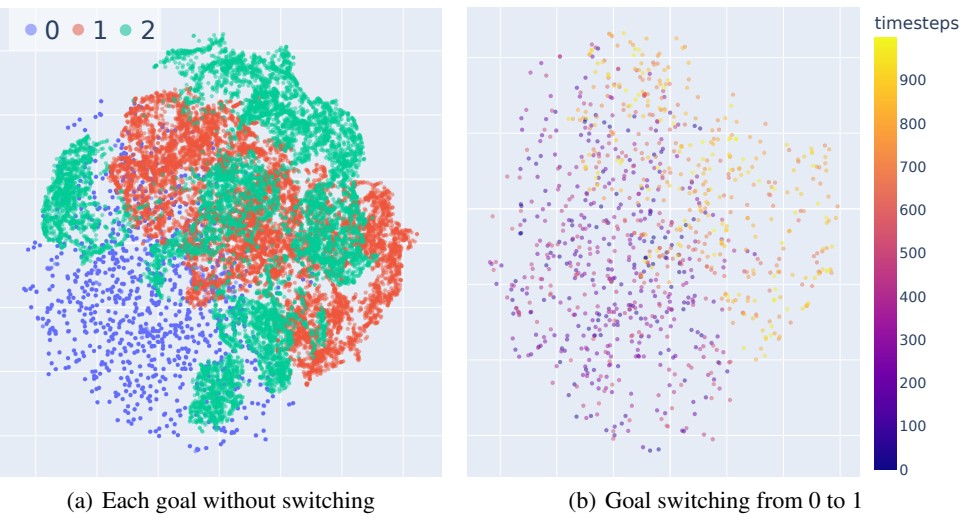

(a) Each goal without switching        (b) Goal switching from 0 to 1

Figure 3: **Left:** The graph represents the state distribution of HalfCheetah for the non-switching task. Legends on the top left corner indicate the goal label. We can observe some overlaps between states on the distributions, which is reasonable as there are overlaps between states in the environments when reaching different goals from a similar set of starting points. **Right:** We visualize the trace of the goal-switching task in terms of the state distribution. The brighter the color, the later the timestep. It demonstrates how states are initially distributed around goal 0 and gradually diffuse to the region of goal 1.

### 3.1.2 ALTERNATIVE ARCHITECTURES

Here we would like to discuss other model architectures and auxiliary tasks for achieving goal switching without demonstration. We initially try to use RvS(states as goals) (Emmons et al., 2021) to achieve arbitrary goal reachability. Referring to Emmons et al. (2021), we use two layer multi-layer perceptron (MLP) with a hidden size of 1024 for this experiment. We also try to use a causal transformer as the backbone, which is the same architecture as ours but replaces the goal label. In this case, the state goal is tokenized following the tokenization scheme for other continuous values. However, we do not find our architecture to provide an apparent performance boost but are only less sensitive to the hyperparameters. Thus, we only report the original results that use

MLP. During training, we sample an arbitrary state from the training dataset at timestep $T = t$, and sample another state that is in the same trajectory at timestep $T > t$. We keep doing this process to train the model until convergence. To aid better strategy learning, we also add some auxiliary tasks. Inspired by Schwarzer et al. (2021), we find that predicting the distance between the state and goal and trajectory labeling prediction works the best. However, by carefully tuning the model, we find that the state-conditioned model can hardly distinguish similar states. It can have decent performance in some tasks but poorly react to the switch task overall. As illustrated in Figure 4, the model can learn different trajectories with a slightly higher variance compared to our model. It also demonstrates how the model performs in goal switching setup. As illustrated, many goal-switching are unsuccessful, showing that this architecture is not capable of this task.

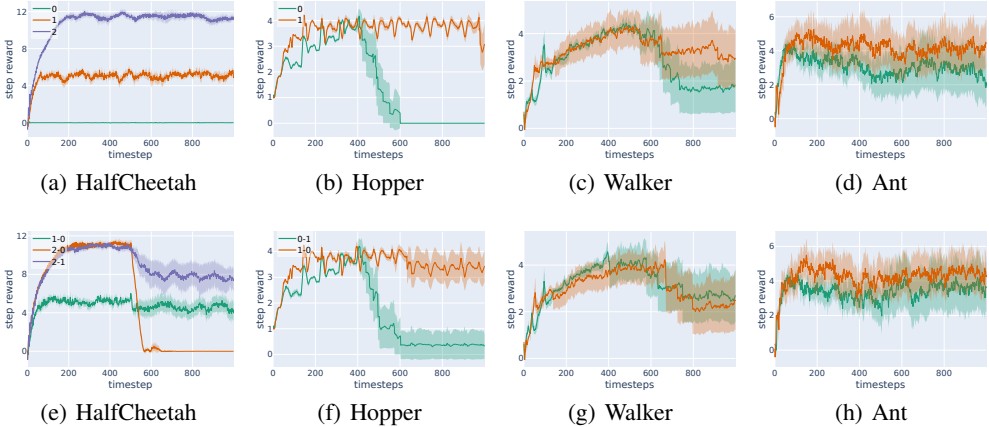

Figure 4: States as goals (a-d) non-switching task (e-h) switching task

We also consider other transformer-based architectures as (Prompt-DT) (Xu et al., 2022) and Decision Transformer Chen et al. (2021). However, Prompt-DT requires sampling a few states from the 'successful' demonstration as prompts to guide the model, which is impossible as the successful demonstration of goal switching is absent from the dataset. Similarly, Decision Transformer requires tweaking directly on the normalized reward, which is also difficult. Hence, a few adjustments are made to their original architectures to suit our task. When composing prompts for Prompt-DT, we replace the old prompt that is toward the old goal and sample states from the new goal label once we switched. For the Decision Transformer, we replace the return-to-goal with the new target at the switch point but subtract the already received return along the way. As illustrated, prompt-DT suffers from all tasks. Whereas Decision Transformer has some good performance when switching to better goals, it does not show competitive performance against GMT. Eventually, we find that if the demonstration of the prompt is changed in the middle of the evaluation for the purpose of goal switching, the model starts to confuse and failed in most of the tests.

## 3.2 ABLATION

### 3.2.1 MASKING RATIO

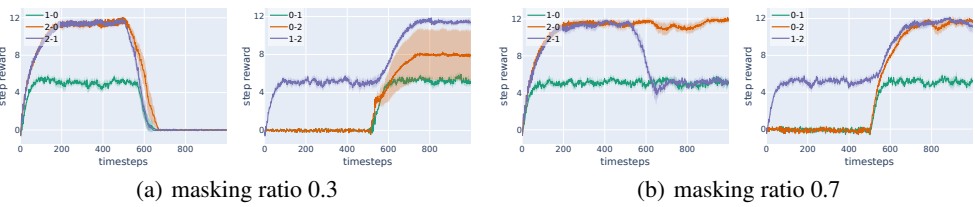

Figure 5: Ablation study of masking ratio in HalfCheetah

Goal masking plays a pivotal role in GMT. In this section, we investigate whether the result is sensitive to the masking ratio. We observe that the masking ratio has no effect on non-switching tasks, which is reasonable. By both increasing or decreasing the masking ratio (as shown in Figure 5), we can observe that various trajectories are no longer switchable to other trajectories or significantly increase in variance. For example, switching from goal label 2 to 1 actually fails when the masking ratio reduces to $0.3$. Also, the models cannot switch to trajectories that are labeled with $0$ when the masking ratio increase to $0.7$. Other tasks demonstrate very similar trends, and we show their results in Appendix A.2.5.

### 3.2.2 CONTEXT LENGTH

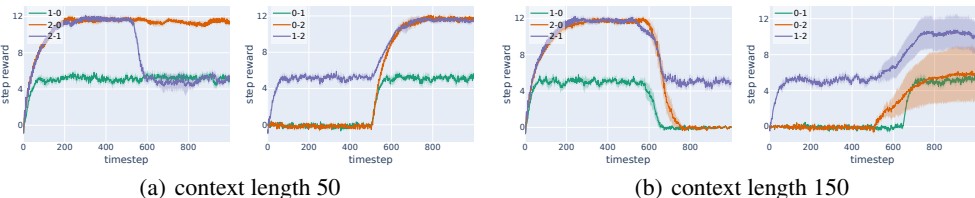

(a) context length 50        (b) context length 150

Figure 6: Ablation study of context length in HalfCheetah

Previous studies (Chen et al., 2021; Xu et al., 2022) found that transformer-based RL algorithms are sensitive to context length. Thus, we demonstrate how context length affects the performance of GMT. As a result, we find that context length only has a limited effect on non-switching tasks. However, too short a context length can cause switching failure, and too long a context length can significantly increase the switch delay and cause higher variance. Other tasks demonstrate remarkably similar trends, and we show their results in Appendix A.2.6.

### 3.2.3 SMOOTHNESS OF TRANSITIONS

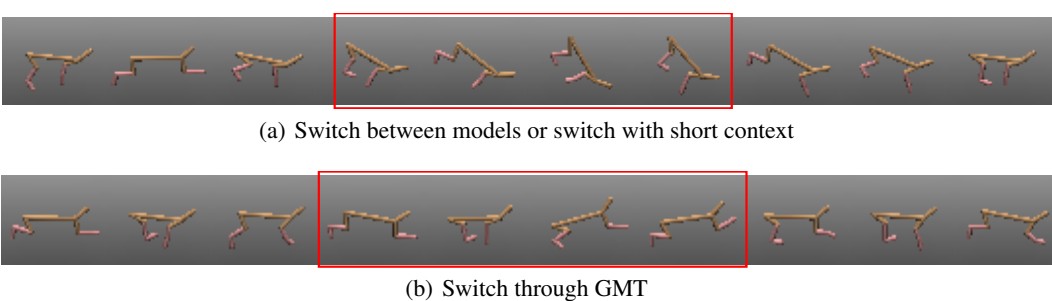

(a) Switch between models or switch with short context

(b) Switch through GMT

Figure 7: Two sequences of frames that renders how HalfCheetah's pose change under different model setup. Every frame on each figure is sampled from every three consecutive frames of the environments. We use two red rectangles to highlight four keyframes where the change happened. As illustrated, goal (pose) switching in (a) is not smooth, showing some strange poses that can hardly be observed at any other time. Whereas, the pose in (b) does not obviously seem to be affected by the goal switching, demonstrating smoother transition and better control.

One intuitive idea is whether we could complete goal-switching tasks by switching between different models. Specifically, we train multiple models for different goals, and then simply switch between different models when goal switching is initiated. We find that the model can switch between different goals but in a different way. Figure 7 illustrates that the models actually restart the process by acting arbitrarily until the model collapses to the state that the model has seen in the training dataset (likely the starting state), which indicates that the model is actually doing things like resetting the environment and then attempting to reach the target state by starting from scratch. Similar trajectories can be also found in GMT with very short context length (i.e. context length $K = 1$). However, GMT performs much smoother transitions between distinct goals, indicating that GMT has barely

been affected by the distribution shifting. We believe that either the distribution shifting or a lack of historical context is to blame.

## 4 RELATED WORK

Goal-conditioned tasks are introduced to reach the desired goal and improve policy control. There has been a lot of research (Andrychowicz et al., 2017; Plappert et al., 2018; Chane-Sane et al., 2021; Chen et al., 2021; Furuta et al., 2021; Ma et al., 2022) on how goal-conditioned approaches can be more effectively generalized to achieve multiple goals or even unseen goals. This work argues that goal-switching can enable better generalizability, especially in the supervised learning setting. Through our method, models can handle unseen transitions, where both goal and state may be observed, but the combination is never seen. Goal-switching can be learned in theory via goal-conditioned reinforcement learning (GCRL) (Liu et al., 2022) by carefully tuning rewards, or goal-conditioned supervised learning (GCSL) (Ding et al., 2019) by feeding sufficient demonstrations. However, in practice, these methods are resource intensive. On the other hand, our proposed approach can infer the switching process between arbitrary goals from an imperfect dataset.

Goal-conditioned tasks can also be formulated as a sequence modeling problem (Chen et al., 2021; Xu et al., 2022; Reed et al., 2022; Furuta et al., 2021). Such sequence models are becoming popular in the RL community recently. New research is being done in a variety of areas, including multi-agents (Wen et al., 2022), planning (Jiang et al., 2022), online learning (Zheng et al., 2022), and offline learning (Chen et al., 2021). We follow recent work (Chen et al., 2021; Furuta et al., 2021) that models the full trajectory of RL using GPT-2-style Transformers, treating it as a sequence modeling problem. Depending on the application circumstance, there are various approaches to modeling trajectories. Our formulation enables sequence modeling to achieve a controllable policy by controlling each step's goal without any reward information. The Categorical Decision Transformer (CDT) proposed in Furuta et al. (2021) is the most similar to our architecture. However, CDT only reformulates the goal learning from a regression problem (as in Decision Transformer) to a classification problem, which is substantially different than what we are trying to achieve.

## 5 CONCLUSION

We present a crucial problem towards controllable policy: the *goal-switching* problem. This problem is challenging for training on fixed datasets, which typically do not cover all the switching processes encountered during deployment. We argue that the existing GCSL methods are not designed to solve such an out-of-distribution (OOD) problem. This paper proposes goal-masked transformers (GMT), a simple but effective GCSL method based on causal transformers. More specifically, we introduce a masking mechanism that forces the model to learn generatively about the connection between goal, state, and action. Additionally, we propose an unsupervised manner to relabel goals from the dataset based on the statics of interest, which would reduce the volatility while switching between goals. We empirically show that GMT can achieve goal-switching with different modal inputs and analyze the impact of various factors on performance.

GMT is a goal-conditioned supervised learning algorithm, which still suffers from the problems of supervised learning, such as heavily relying on the dataset's quality. Switching between goals can be done smoothly if the dataset quality is good, but the switching process may have a higher variance if the dataset quality is terrible. Meanwhile, a high-quality dataset allows GMT to learn better and more complex policies. GMT is not limited to the GCSL paradigm and can be easily extended to offline RL or GCRL in conjunction with existing works.

Furthermore, GMT does not learn the goal-switching controller that determines when to change the goal. We emphasize that the problem GMT is trying to solve is the ability to learn goal-switching from imperfect datasets. This dramatically enhances the controllability of the policy and enriches human-computer interaction (HCI). At the same time, GMT makes it straightforward to use text tokens as goal tokens, significantly improving the potential of HCI by fusing them with natural language. We leave these as future works.

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

# A APPENDIX

## A.1 MODEL

### A.1.1 TOKENIZATION

The trajectories may include data from multiple modalities. We transform all data into a flat sequence of tokens in order to enable the processing of this multi-modal data. For GMT, we apply the following tokenization scheme to produce better results.

- Continuous values (e.g. joint positions of locomotion tasks) are normalized to a standard normal distribution. The mean and variance that are used for normalization are calculated separately on each dataset.

- Discrete values (e.g. Atari button presses) are used as integer tokens directly without further processing.

- Images (e.g. Atari raw observations) are normalized between $[0, 1]$ for each pixel (e.g., divided by 255 for Atari games). Inspired by Multi-Game Decision Transformers (MGDT) (Lee et al., 2022), we experiment with splitting images into small patches, but find it drastically reduces the performance due to the much shorter receptive fields of the model and therefore we abandon the change.

Data is transformed into tokens, which are sorted in the order of goal, state, and action. Each agent episode is comprised of multiple agent timesteps in time order.

### A.1.2 IMPLEMENTATION DETAILS

Table 2: Hyper-parameters

| Env. | Hidden size | Blocks | Heads | Context length |
|---|---|---|---|---|
| MuJoCo | 128 | 4 | 2 | 100 |
| Atari | 768 | 4 | 12 | 50 |

We use a unified architecture for both Atari games and MuJoCo tasks, the hyper-parameters are listed in Table 2. Blocks represent the number of multi-head self-attention (Vaswani et al., 2017) blocks we have for the model. During training, we only calculate the action loss and goal loss. Regardless of the goal embedding provided to the model (either true token or $[mask]$ token), we calculate the loss towards the true label as described in Section 2.2. Training takes approximately an hour on a single NVIDIA A100 GPU for each MuJoCo task and approximately two days for each Atari game on eight NVIDIA A100 GPUs.

### A.1.3 DEPLOYMENT

During deployment, we first select a goal label as described in the previous section. The first observation is then produced by the environment and tokenized before being added to the sequence. GMT autoregressively samples the action vector one token at a time. This action is sent to the environment which steps and yields a new observation. The above procedure repeats. Each transition is made up of a goal, state, and action. All past observations and actions are constantly visible to GMT in its context window of $K$ tokens. The goal can be changed at any time during deployment, simply by feeding the model with the desired goal token.

Motivated by MGDT, we introduce an inference-time probability-based method, called expert action inference, to sample actions in Atari games. Rather than always selecting the action with the highest logits (by using argmax), we sample the action with the top $q$ quantile. Then, we sample actions based on the normalized logits of selected actions. In practice, we find $q = 0.6$ to work the best. As otherwise mentioned, we use this action sampling strategy by default in Atari games.

It is important to note that all the methods introduced in this section are only applied in the inference procedure of GMT. During training, the goals in each trajectory remain the same and cannot be altered while training. Expert action inference is likewise only used during evaluation for Atari.

## A.2 EXPERIMENTS

### A.2.1 MUJOCO

Table 3: Mujoco dataset statistics v.s. evaluation performance without any switching for each dataset label. *gl* stands for *goal label*.

|  | HalfCheetah | | | Hopper | | Walker | | Ant | |
|---|---|---|---|---|---|---|---|---|---|
|  | $gl = 0$ | $gl = 1$ | $gl = 2$ | $gl = 0$ | $gl = 1$ | $gl = 0$ | $gl = 1$ | $gl = 0$ | $gl = 1$ |
| Dataset | $-286 \pm 79$ | $4775 \pm 79$ | $10670 \pm 230$ | $1430 \pm 124$ | $3594 \pm 35$ | $2006 \pm 218$ | $4921 \pm 31$ | $987 \pm 342$ | $5141 \pm 58$ |
| Evaluation | $-0.31 \pm 0.60$ | $4940 \pm 146$ | $10747 \pm 119$ | $1626 \pm 285$ | $3394 \pm 678$ | $2551 \pm 729$ | $4911 \pm 30$ | $4067 \pm 363$ | $5203 \pm 91$ |
| Traj. Num | 500.00 | 363.00 | 500.00 | 300.00 | 300.00 | 158.00 | 300.00 | 195.0 | 300.00 |

Table 3 illustrates that our model can learn the representation of trajectories with different labels. However, there are two exceptions which are $gl = 0$ for HalfCheetah and $gl = 0$ for Ant. We realize that there is a noticeable gap between the mean value for these tasks. By further investigating, we find that this may be caused by differences in reward calculations between the environment that perform data collection and evaluations. We decide to live with this difference and use this mean as the new standard to determine whether or not the switch task is successful.

### A.2.2 ATARI

We select one Atari game (Breakout) to demonstrate that GMT can also be applied to video games with image inputs and complex policies. For each Atari game, the chosen statistics of interest is the reward. We perform various data augmentations for Atari games during training, including random crop and random rotation.

We cluster the Breakout into 11 goals. Figure 10 shows the trajectories sampled from each goal label in the Breakout game. We can observe several patterns across different trajectory labels. For example, goals 0 and 10 tend to focus on breaking the middle part, however, 2 and 5 tend to break evenly without too much preference. Additionally, the agent tends to break more blocks on the top as the value of the goal label increase. This is a reasonable behavior as breaking those blocks on the top tends to give higher rewards. It is possible to reduce this variance by clustering trajectory using the normalized score which only considers the sign of the score as in previous work (Mnih et al., 2015). However, this change is not adopted because the normalized reward in this scenario depends on how many blocks the agent breaks, which is not the statistic of interest for this experiment.

As there are too many combinations for switching, we illustrate a few here (specifically between 2, 5, and 10) to evaluate the ability of GMT. As demonstrated in Figure 8 we expect the total reward of the trajectory sits around $23 \times 0.3 + 74 \times 0.7 = 57.5$ when switching from 2 to 10 at timestep 300, and sits around $74 \times 0.3 + 23 \times 0.7 = 38.3$ when starting with 10 and end with 2 under the same procedure. We use $0.3$ and $0.7$ as each part generally takes 30% and 70% of the entire trajectories, respectively. By using a similar method, we can also calculate the expected return when switching between 2 and 5. They are 35.1 and 25.9, for goal $2 \rightarrow 5$ and $5 \rightarrow 2$, respectively. We observe that switches are successful across all the illustrated examples, and the final episodic reward is close to our expectations. We also found that the expectation calculation tends to overestimate the total reward that an agent can receive when switching from a better performance goal to the worse one (e.g. $10 \rightarrow 2$ as we demonstrated). Unlike the MuJoCo task, this is a reasonable phenomenon in Breakout as the block closer to the top has high marks, which can only be hit at the later game. Thus, switching to a better strategy in the later game tends to receive more rewards. This can also be observed from the left line plot, in which the gradient of the lines is much high in later games.

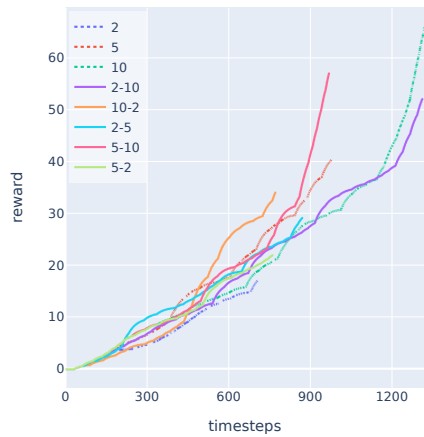 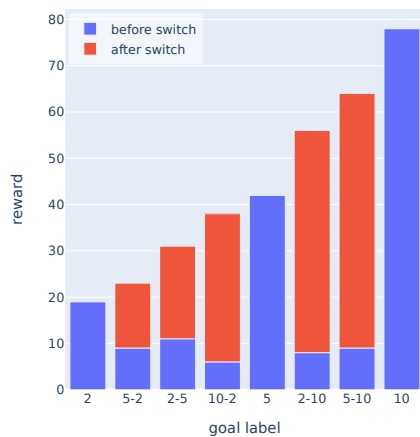

Figure 8: **Left:** The rewards trends within an episode that the agent gets for different goal labels with or without switching. We use dash lines to represent cases without switching as references. All switches are happened at timestep 300 by gradually replacing the old goal with the new goal to provide agents with some context before and after the switch. For better illustration, all lines are smoothed by using an exponential running average with smoothed parameter $\beta = 0.97$. **Right:** We try to emphasize the total reward before and after the reward is changed in this figure. Bars with multiple colors represent the total reward received in different stages, whereas bars with only one color (blue) mean that goals are consistent across the episode without changes.

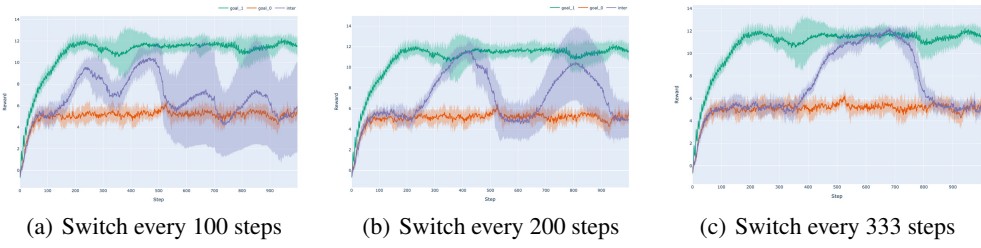

(a) Switch every 100 steps     (b) Switch every 200 steps     (c) Switch every 333 steps

Figure 9: GMT switch more than once within an episode

### A.2.3 MORE SWITCH WITHIN ONE EPISODE

It is clear that our modal is capable of switching more than once within an episode as shown in 9. When GMT switches every 100 steps, 15/15 complete the first switch. However, only 3/15 still on the track at the end (success all 9 switches). When GMT switches every 200 steps, 13/15 can finally made it (success all 4 switches). When GMT switches every 300 steps, all attempts succeed (success all 4 switches). Apparently, the more frequent the switch, the higher variance and failure rate as the switch happens before the finish of another switch. This is because all middle transitions are 'guessed' by our model. Keep staying and operating in such state space will eventually lead to unrecoverable error. However, our model still have decent performance when the interval is reasonable.

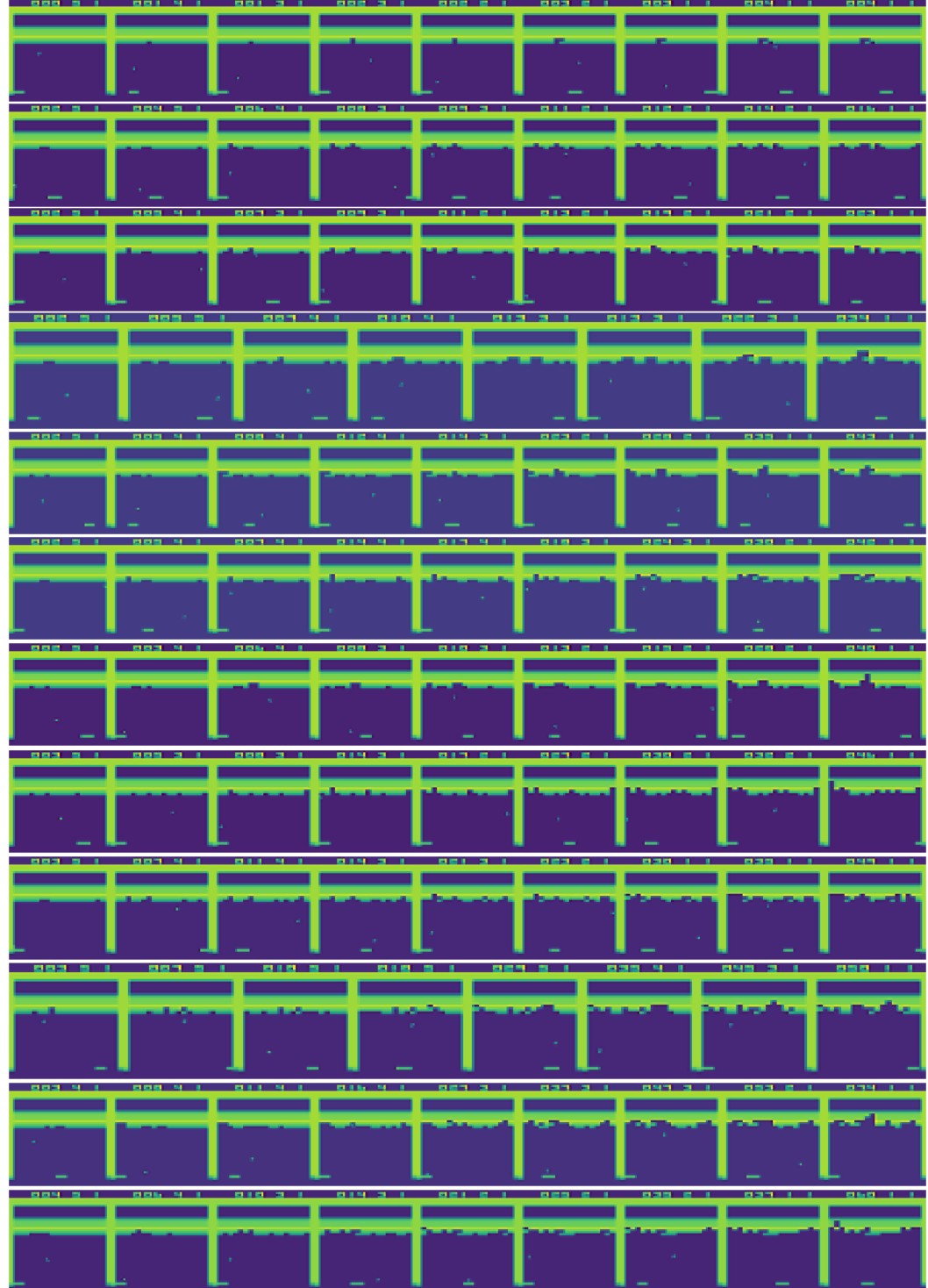

Figure 10: Breakout trajectory samples. From top to down are goals labeled from 0 to 10.

### A.2.4 STATE DISTRIBUTIONS FOR MORE ENVIRONMENTS

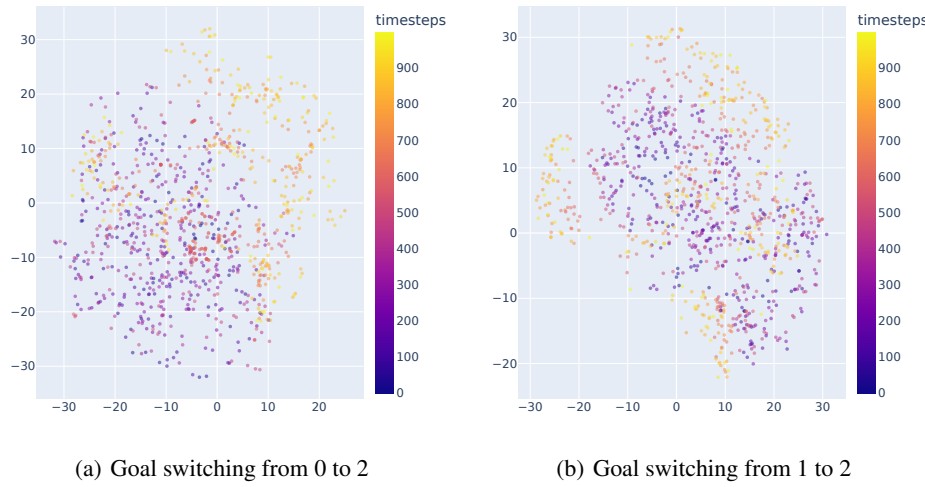

(a) Goal switching from 0 to 2

(b) Goal switching from 1 to 2

Figure 11: State distribution in HalfCheetah

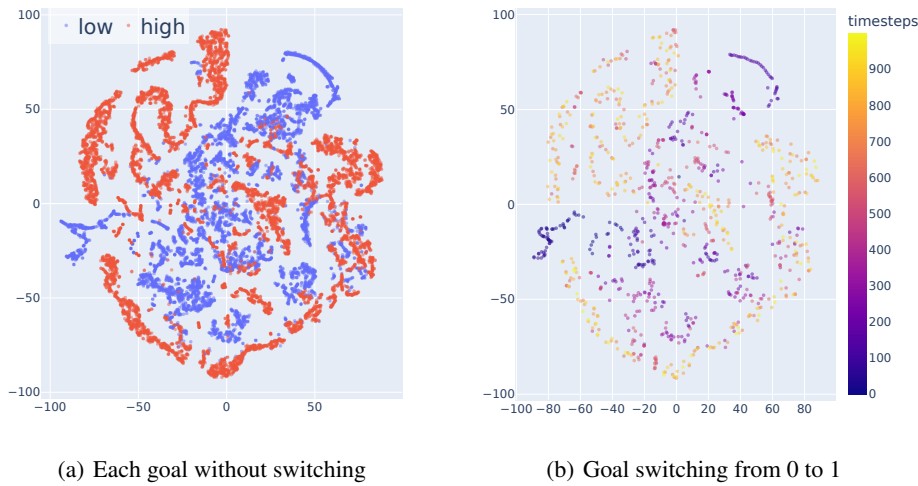

(a) Each goal without switching

(b) Goal switching from 0 to 1

Figure 12: State distribution in Walker

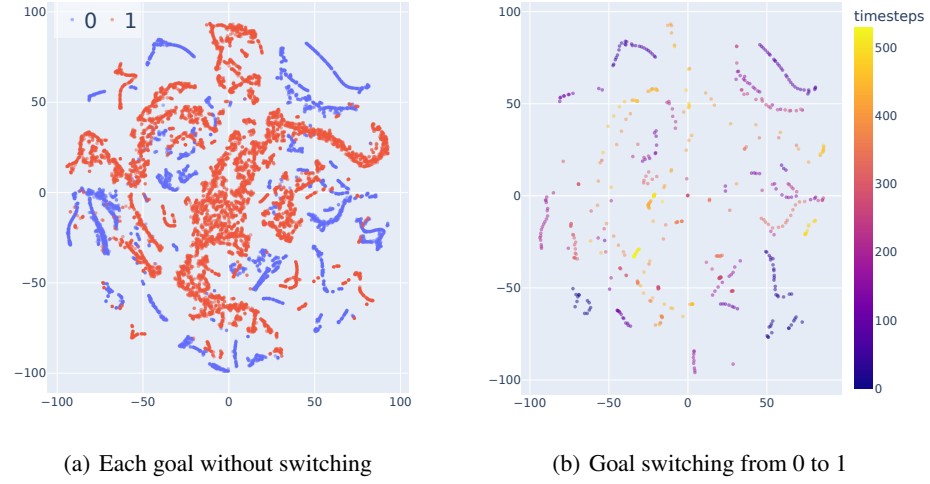

(a) Each goal without switching    (b) Goal switching from 0 to 1

Figure 13: State distribution in Hopper

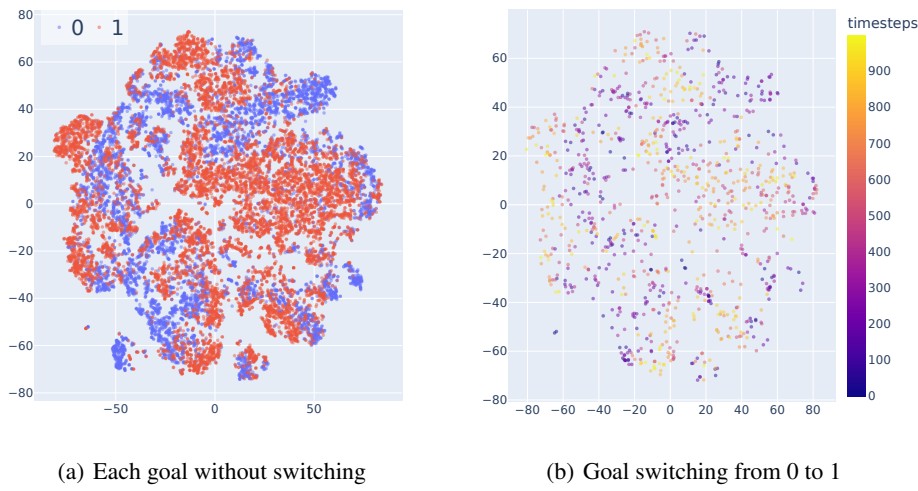

(a) Each goal without switching    (b) Goal switching from 0 to 1

Figure 14: State distribution in Ant

### A.2.5    MASKING RATIO FOR MORE ENVIRONMENTS

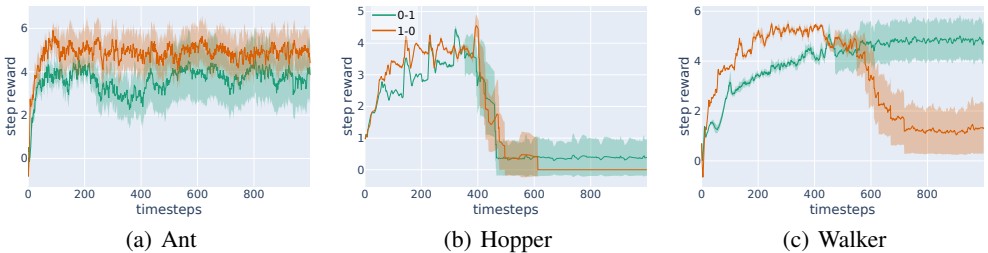

|  (a) Ant  |  (b) Hopper  |  (c) Walker  |

Figure 15: Ablation study of masking ratio with 0.3 on three different tasks with switching

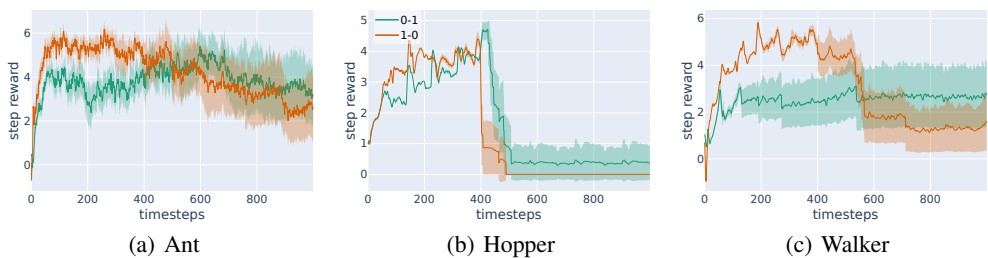

|  (a) Ant  |  (b) Hopper  |  (c) Walker  |

Figure 16: Ablation study of masking ratio with 0.7 on three different tasks with switching

### A.2.6    CONTEXT LENGTH FOR MORE ENVIRONMENTS

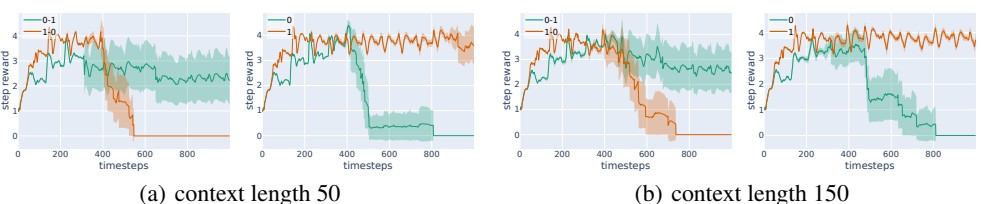

|  (a) context length 50  |  (b) context length 150  |

Figure 17: Ablation study of context length in Hopper

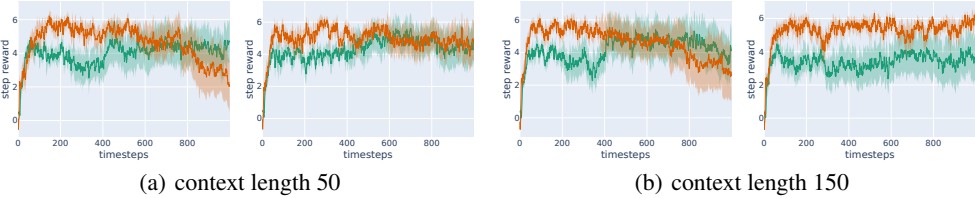

(a) context length 50          (b) context length 150

Figure 18: Ablation study of context length in Ant

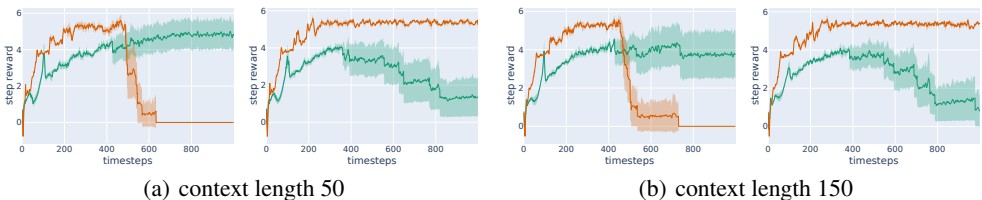

(a) context length 50          (b) context length 150

Figure 19: Ablation study of context length in Walker

### A.2.7 FINER-GRAINED GOAL CLUSTERING

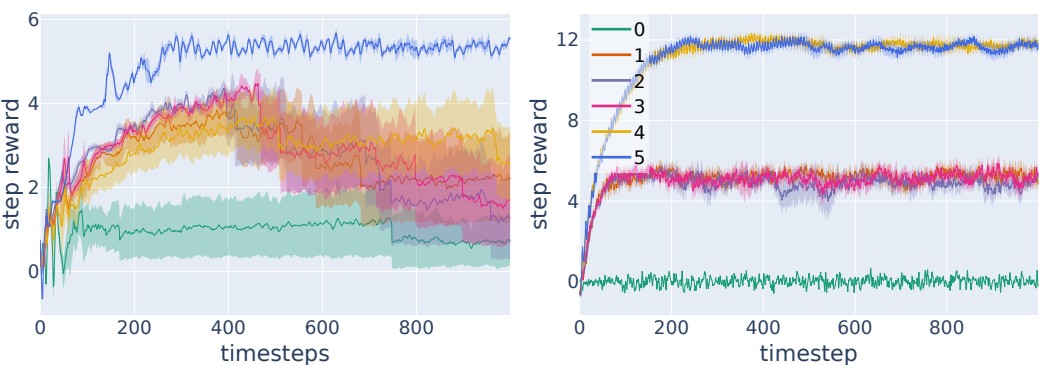

Figure 20: Walker with 6 goal labels        Figure 21: HalfCheetah with 6 goal labels

Both Figure 20 and 21 shows that too many class labels will eventually result in goals that are unable to distinguish between each other. Many of the trajectories overlap with each other in both environments.

