# OpenReview forum: "Towards Controllable Policy through Goal-Masked Transformers"
_ICLR.cc/2023/Conference — Submitted to ICLR 2023_

### Official Review · Reviewer_33J4 · 2022-10-24

**Confidence:** 5
**Correctness:** 2
**Technical Novelty And Significance:** 2
**Empirical Novelty And Significance:** 2
**Recommendation:** 3

**Clarity, Quality, Novelty And Reproducibility:**

### Quality
There are some points that should be improved in writing:
- The reason why trajectory masking doesn't work well is not explained in Section 2.3 (recursively referenced in footnote 2).
- How many random seeds did you use? I only found the description of  "15 times" rollout, for the evaluation.
- Please use `\citep` in the second paragraph of Section 4.

### Clarity
There are some unclear points in the paper:
- The definition of **goal** is missing, although "goal" variable of this paper seems different from typical goal-conditioned RL papers.
- The is no architectural detail. I guess most parts are borrowed from Decision Transformer or Generalized Decision Transformer, but there are no descriptions or citations.
- The definition of $\mathcal{S_k},\mathcal{A_k}, \mathcal{G_k}$ in Section 3.1.1 is unclear.
- The reason why step reward is an evaluation metric is unclear, while x-velocity in the trajectory seems to be related to goal variable.


### Originality
Since the architecture seems to be on top of Decision Transformer, and the problem formulation is similar to Generalized Decision Transformer, the originality of this paper can be limited (random goal-masking and auxiliary goal prediction loss).

**Strength And Weaknesses:**

### Strength
- The writing is easy to follow.
- The learned transformer policy seems to smoothly switch the goal during a single episode, compared to multiple models or shot context (Figure 7).


### Weaknesses
- There are no baselines for comparison. Decision Transformer [1], Categorical Decision Transformer or Bi-directional Decision Transformer [2],  Prompt Decision Transformer [3], GoFAR [4], WGCSL [5], and LiPS [6] can be relevant methods to be compared (of course, not the all but more is better).
- The goals in MuJoCo and goals in Atari are quite unclear. In MuJoCo, goals seem to stand for the discrete label that is clustered with k-means (based on x-velocity in the trajectory). In Atari, goals seem to stand for the returns in the trajectory. As far as I know, those abstract quantities are often not considered as a goal in the existing goal-based RL literature.
- The goal switching is limited to once in the episode. Also, the number of goals is limited to 3 (0, 1, 2 in MuJoCo, 2, 5, 10 in Atari).


[1] https://arxiv.org/abs/2106.01345

[2] https://arxiv.org/abs/2111.10364

[3] https://arxiv.org/abs/2206.13499

[4] https://arxiv.org/abs/2206.03023

[5] https://arxiv.org/abs/2202.04478

[6] https://arxiv.org/abs/2012.03548


**Summary Of The Paper:**

This paper studies offline goal-conditioned reinforcement learning problems and proposes a novel offline goal-conditioned supervised learning method, Goal-Masked Transformers (GMT), which applies random goal-masking and predict goals as an auxiliary tasks. The learned policy could successfully switch the goals during a single episode (e.g. goal 0 --> goal 1) in both MuJoCo or Atari environments

**Summary Of The Review:**

This paper lacks the definition of goals, proper comparison against relevant baselines, and novelty compared to Decision Transformer or Generalized Decision Transformer. Considering those aspects, I vote for rejection.

---

> ### Author Response · Authors · 2022-11-18
> **Responds to reviewer 33J4**
>
> We thank the reviewer for the valuable comments and careful reading. We address all the concerns as follows.
>
> > There are no baselines for comparison. Decision Transformer [1], Categorical Decision Transformer or Bi-directional Decision Transformer [2], Prompt Decision Transformer [3], GoFAR [4], WGCSL [5], and LiPS [6] can be relevant methods to be compared (of course, not the all but more is better).
>
> We move part of the results and explanations from the appendix to the section. We further put a table to summarize the performance of baselines (including RvS, Prompt-DT, and Decision Transformer) on our problem. We illustrate the final results in success rate at the beginning of Section 3. A few adjustments are made towards each baseline as their original architecture cannot directly be applied to our tasks (Section 3.1.2). In conclusion, none of them show comparable performance in goal-switching problems against GMT.
>
> > The goals in MuJoCo and goals in Atari are quite unclear. In MuJoCo, goals seem to stand for the discrete label that is clustered with k-means (based on x-velocity in the trajectory). In Atari, goals seem to stand for the returns in the trajectory. As far as I know, those abstract quantities are often not considered as a goal in the existing goal-based RL literature.
>
> and
>
> > The definition of goal is missing, although "goal" variable of this paper seems different from typical goal-conditioned RL papers.
>
> We define the goal as statistics of interests, for example, reward, speed, number of blocks that break in a game, the dynamics of robots, and strength of game bots. These statistics are generally not provided in the demonstration but are implicitly included in the data. The statistics can be linked to each trajectory by some kind of mapping function. Namely, we can mathematically formulate the goal as $g = f(s_0^i, a_0^i, s_1^i, a_1^i,..., s_{T_i-1}^i, a_{T_i-1}^i;e)$ where $e$ indicates the statistics of an interest and $g$ indicates the goal. Given the statistics of a specific interest $e$, the function $f$ can map the trajectory into the corresponding goal.
>
> We updated the definition in Section 2 (Method), which can be found in the reversion.
>
>
> > The goal switching is limited to once in the episode.
>
> We only demonstrate switching within an episode since it is difficult. However, it is not limited to one. We update the paper to show the results of multiple switches within an episode (see A.2.2). Switching between episodes is equivalent to resetting the environment and starting from the beginning, which is trivial.
>
> > Also, the number of goals is limited to 3 (0, 1, 2 in MuJoCo, 2, 5, 10 in Atari).
>
> We experiment with more goals for MuJoCo environments and put the results in A.2.7. The result is coherent with our claim. The total number of goals in Atari games is 10. We use 2, 5, and 10 to demonstrate the effect of the switch in Atari, as the gaps between these goals are large enough to show clear trends. Other combinations are also possible, but it is only possible to show some goal combinations within the paper as there are too many.
>
> > The reason why step reward is an evaluation metric is unclear, while x-velocity in the trajectory seems to be related to goal variable.
>
> The step-based plot is also x-velocity for MuJoCo. We used the total reward received for Atari tasks until the switch point or end.
>
> > The definition of symbols in Section 3.1.1 is unclear.
>
> Thanks for the suggestion. We remove the duplicated definition of terms and keep them coherent with what we defined in the Method section.

---

> > ### Comment · Reviewer_33J4 · 2022-11-29
> > **Response**
> >
> > I thank the author for the update of your manuscripts and detailed response. The paper seems better than before, but I still have a concern about the novelty; the formulation of "goal" in this paper and goal-switching seems to be very relevant to Generalized Decision Transformer, but it is not explicitly stated. Also, I am still confused about the definition of "goal" in Atari experiments. I don't think it is justified enough. Besides, Table 1 doesn't provide a variance of success rate. For the reasons above, I'd like to maintain my rating as it is.

---

### Official Review · Reviewer_HqN1 · 2022-10-25

**Confidence:** 4
**Correctness:** 2
**Technical Novelty And Significance:** 2
**Empirical Novelty And Significance:** 2
**Recommendation:** 3

**Clarity, Quality, Novelty And Reproducibility:**

As mentioned in the strengths and weaknesses section, this paper is novel but there are a lot of details missing about 1) why the approach is important (why goal-switching is difficult), 2) how the approach works, and 3) lack of baselines and evaluations on the right metrics.

It seems like the approach is reproducible.

**Strength And Weaknesses:**

Strengths:

- To my knowledge, the idea of goal masking in GCSL-style methods is novel
- The method has interesting promise and good applications

Weaknesses

In my opinion, a major weakness is the lack of detail around why certain design decisions are made about this approach. For example, I think more discussion is needed as to why goal switching is a difficult problem with the current framework. Wouldn't since we learn a policy pi(a_t | g_t, a_t) via hindsight relabelling, shouldn't the policy automatically react well to changing g_t. Another detail missing is how exactly the training data is selected, how the goals are chosen. Couldn't we think of goal-switching as two runs of shorter length?

Secondly, I think more evaluations are needed. There are no comparisons to baseline approaches (Decision Transformer, standard GCSL, etc). It is important to show how these methods perform when the goal is switched in the middle of the episode. The metric used for the Mujoco tasks is the episode reward, but for goal-conditioned frameworks, the success rate is what is important. Can the policy reach the goal? It is unclear why clustering is needed, adds a strong assumption that we need to know the full state space, which will not be true when scaling to more difficult settings, thus I think this approach should be tried on random (not clustered goals). Ablations such as no-masking, are also missing.




**Summary Of The Paper:**

The goal of this paper is to try to learn a goal-conditioned policy which is robust to goal-switching, i.e. when the goal given to the policy changes in the middle of the episode. The proposed approach uses a causal transformer, and proposes masking the goal randomly to ensure that the model (GMT) is able to predict the goal from action and state information. The paper presents results on MuJoCo and Atari environments.

**Summary Of The Review:**

Overall, this paper presents an interesting direction and a novel approach that provides an initial result towards this direction, but design choices and explanations of the approach are unclear. Additionally, this approach needs a more thorough set of evaluations (see strengths and weaknesses sections).

---

> ### Author Response · Authors · 2022-11-18
> **Part 1 Responds to Reviewer HqN1**
>
> We thank the reviewer for the positive comments and the recognition of our work. Concerns mainly fall on design decisions and comparison against other baselines. We address all the points as follows.
>
> >In my opinion, a major weakness is the lack of detail around why certain design decisions are made about this approach. For example, I think more discussion is needed as to why goal switching is a difficult problem with the current framework.Wouldn't since we learn a policy pi(a_t | g_t, a_t) via hindsight relabelling, shouldn't the policy automatically react well to changing g_t.
>
> If the goal here represents a state, we primarily believe that learning this transition $\pi(a_t | g_t, a_t)$ can help with the goal-reaching and goal-switching problem, as stated in [1]. However, our experiments show that this is not always true. We discussed the experimental result in 3.1.2. To summarize, we closely follow the setup mentioned in the paper, even adding extra auxiliary tasks to help them work better in our problem setting. The results show that the model can reach a fixed target with slightly higher variance than our method. However, as illustrated in Figure 13 (e->h), almost all switch tasks fail. We discussed why this is the case in the paper. Generally, data from D4RL are trained by a typical RL algorithm that tries to maximize its return. It tries to bump up the curve from bottom to up, as illustrated on all our plots, regardless of reaching a value high value (expert demonstration), medium values (medium demonstration), or failure (random demonstration). Thus, learning $\pi(a_t | g_t, a_t)$ does not guarantee the model can generalize to unseen transition mode, for example, switching the model from goal label 3 to goal label 1 in our experiments.
>
> If the goal represents the same goal we proposed in the literature, we did some ablation studies on context length in section 3.2.2. Directly learning $\pi(a_t | g_t, a_t)$ is equivalent to use context length = 1. However, as illustrated in Figure 6, the model struggle to switch from 2 -> 0 and 1 -> 0 by only halving the context length. Further reducing context length leads to even worse results.
>
> > Another detail missing is how exactly the training data is selected, how the goals are chosen.
>
> It is provided in Section 2.1 under the Method section.
>
> For MuJoCo task
>
> "We conduct 4 MuJoCo (Todorov et al., 2012) locomotion environments with continuous
> control from the D4RL (Fu et al., 2020) benchmark, including HalfCheetah, Hopper, Walker, and
> Ant. The locomotion task is the fundamental task in the field of robotics and we can show the control of the movement speed (along the x-axis) by switching target poses. We combine the Medium and Medium-Expert datasets provided in the D4RL benchmark as the training dataset for each locomotion environment except for HalfCheetah. On the HalfCheetah task, we also add the Random dataset where trajectories are totally generated by a random policy."
>
> For Atari task
> "We demonstrate that GMT can also be applied to games, such as video games. The RLUnplugged (Gulcehre et al., 2020) dataset provides trajectories of numerous Atari games (Bellemare et al., 2013) collected from the training progress of a DQN (Mnih et al., 2013) agent. The dataset covers various demonstrations, including agent behavior at all learning stages. We train GMT over the Breakout dataset to show that GMT can switch between different levels of policies according to the changes in goal."
>
> For how to select the goal
> "Here we propose an unsupervised manner to aggregate and label trajectories for the aforementioned offline datasets. We first perform a k-means clustering (Lloyd, 1982) along the statistics of interest (e.g. we are interested in the speed of locomotion on the x-axis in MuJoCo) with a range of k, and then calculate the silhouette score (Rousseeuw, 1987) of each cluster. As there are now goodness measurements of clustering with different k, we sample a k based on the normalized silhouette score from the top 85th percentile."

---

> > ### Author Response · Authors · 2022-11-18
> > **Part 2 Responds to Reviewer HqN1**
> >
> > > Couldn't we think of goal-switching as two runs of shorter length?
> >
> > We examine this possibility in Section 3.3 in the revision by switching between multiple trained models targeting different goals. However, the transition quality greatly suffered. As mentioned above, the most challenging part of the goal-switching problem is how the model can generalize to unseen transitions. For example, the switch between high speed and medium speed in the HalfCheetah example as offline data only demonstrates how to reach high speed while standing without knowing how to start from very high but reach a "worse" state (the medium speed).
> >
> > > Secondly, I think more evaluations are needed. There are no comparisons to baseline approaches (Decision Transformer, standard GCSL, etc). It is important to show how these methods perform when the goal is switched in the middle of the episode.
> >
> > We move part of the results and explanations from the appendix to the section. We further put a table to summarize the performance of baselines (including RvS, Prompt-DT, and Decision Transformer) on our problem. We illustrate the final results in success rate at the beginning of Section 3. A few adjustments are made towards each baseline as their original architecture cannot directly be applied to our tasks (Section 3.1.2). In conclusion, none of them show comparable performance in goal-switching problems against GMT.
> >
> > >The metric used for the Mujoco tasks is the episode reward, but for goal-conditioned frameworks, the success rate is what is important. Can the policy reach the goal?
> >
> > Thanks for the valuable feedback. We update the paper by adding success rates on each task to better demonstrate the results alongside the current plots.
> >
> > >It is unclear why clustering is needed, adds a strong assumption that we need to know the full state space, which will not be true when scaling to more difficult settings, thus I think this approach should be tried on random (not clustered goals).
> >
> > The clustering procedure only performs on offline data. Accessing all states in the data is straightforward. However, accessing the full state space of the problem is impossible as offline tasks can only obtain knowledge from the limited and fixed dataset. Also, offline data quality can sometimes be terrible for some tasks (e.g., Ant). Without clustering, each trajectory is most likely regarded as having different goals. However, we demonstrate that an unreasonable increase in clusters can harm performance (as shown in A.2.7) as there is almost no difference between trajectories, for example, received 10 (as episode reward) and 10.1 from the goal perspective. Thus, we need a strategy to measure the similarity across different trajectories and group those with similar properties. This allows us to further force the model (GMT) to interact with coarse-grained training data, which mitigates the issue.
> >
> > >Ablations such as no-masking, are also missing.
> >
> > Thanks for the valuable feedback. The results of our model without masking are now listed in Section 3, Table 1.
> >
> > Reference:
> >
> > [1] Emmons, Scott, et al. "RvS: What is Essential for Offline RL via Supervised Learning?." arXiv preprint arXiv:2112.10751 (2021).

---

### Official Review · Reviewer_S5ck · 2022-10-27

**Confidence:** 5
**Correctness:** 2
**Technical Novelty And Significance:** 2
**Empirical Novelty And Significance:** 2
**Recommendation:** 1

**Clarity, Quality, Novelty And Reproducibility:**

The paper is mostly well written. The description of the goal switching problem can be improved, since the problem of goal switching is essentially the problem of achieving multiple goals in the term of goal conditioned RL.


**Strength And Weaknesses:**

I think this paper presents an interesting idea of training goal conditioned policies. However, I do have some important concerns about the proposed method.


I’m not convinced about the novelty of the proposed method. The proposed method is essential decision transformers [1] applied to goal conditioned reinforcement learning, where the return-to-go is replaced by the goal in the transitions, where the goal can be considered a special case of rewards. Hence the only novelty for the proposed method is the application of masking in goals, which I do not believe is enough contribution.

My major concern about this paper is that the authors have not included any comparison of the proposed method to previously proposed baselines. Goal or return conditioned RL is a relatively mature subfield in RL with many prior works. Just to name a few, I believe the specific tasks solved in the paper can also be solved by prior methods such as Decision Transformers [1], Trajectory Transformers [2], HER [3], GCSL [4], RCP [5] and RvS [6]. Therefore, it is impossible to verify the empirical performance of the proposed method without comparing to some of these baselines.


### References
[1] Chen, Lili, et al. "Decision transformer: Reinforcement learning via sequence modeling." Advances in neural information processing systems 34 (2021): 15084-15097.

[2] Janner, Michael, Qiyang Li, and Sergey Levine. "Reinforcement learning as one big sequence modeling problem." ICML 2021 Workshop on Unsupervised Reinforcement Learning. 2021.

[3] Andrychowicz, Marcin, et al. "Hindsight experience replay." Advances in neural information processing systems 30 (2017).

[4] Ghosh, Dibya, et al. "Learning to reach goals via iterated supervised learning." arXiv preprint arXiv:1912.06088 (2019).

[5] Kumar, Aviral, Xue Bin Peng, and Sergey Levine. "Reward-conditioned policies." arXiv preprint arXiv:1912.13465 (2019).

[6] Emmons, Scott, et al. "RvS: What is Essential for Offline RL via Supervised Learning?." arXiv preprint arXiv:2112.10751 (2021).


**Summary Of The Paper:**

The paper focuses on the problem of goal conditioned reinforcement learning, where the authors propose a method to train goal conditioned policies via supervised learning using Transformers. Specifically, the authors treat the transitions of state, action and goal in a trajectory as a sequence, and train a Transformer network with causal attention to predict the future action and goals in an autoregressive way. In addition, in order to make the model pay more attention to goals, the authors also introduce the strategy of goal masking during training, where the ground truth goals are replaced with a mask token with certain probability.

The authors evaluate the proposed method in Gym locomotion and Atari tasks, and the results show that the proposed method is able to perform the correct task condition on given goals.


**Summary Of The Review:**

Given the concerns about novelty and empirical evaluations, I cannot recommend acceptance of the paper in its current state.

---

> ### Author Response · Authors · 2022-11-18
> **Responds to reviewer S5ck**
>
> We thank the reviewer for the valuable comments. The comments focus mostly on the lack of novelty and no clear comparison with baseline methods. We address all the points as follows.
>
> >I’m not convinced about the novelty of the proposed method. The proposed method is essential decision transformers [1] applied to goal conditioned reinforcement learning, where the return-to-go is replaced by the goal in the transitions, where the goal can be considered a special case of rewards. Hence the only novelty for the proposed method is the application of masking in goals, which I do not believe is enough contribution.
>
> We respectfully disagree that our goal is a special case of rewards. Instead, the goal in our model is generalized statistics of interests, as we defined in the literature, which can utilize the reward information that can be fairly easily obtained in offline RL datasets. Thus, we believe that the reward is a special case of our goal.
>
> Considering the return-to-go, it is the most significant difference between conventional goal-conditioned RL approaches and Decision Transformer-like methods. The key idea is to condition the model's output with the expected future reward that the model can receive until the end of the episode. Hence, methods with similar architecture to the Decision Transformer update the goal for each interaction with the environment, regardless of whether the environment's feedback is directly utilized (e.g., Decision Transformer) or not (e.g., Bi-directional Decision Transformer, Multi-Game Decision Transformer). However, the goal in our model is invariant during the interaction under the same task setting as Decision Transformer (unless manually changed).
>
> Besides, the meaning of our goal is substantially different from other rewards or return-to-go. They are abstract representations of the target trajectories or behavior that can guide the model to achieve specific goals. This design further enables fine-grained control over the behavior of the policy, compared to tweaking the normalized reward in Decision Transfomer is neither efficient nor intuitive. Other methods (e.g., Bi-directional Decision Transformer, Multi-Game Decision Transformer) that use purely autoregressive goals or learned goals can exacerbate this issue.
>
> We can mathematically formulate the goal as $g = f(s_0^i, a_0^i, s_1^i, a_1^i,..., s_{T_i-1}^i, a_{T_i-1}^i;e)$ where $e$ indicates the statistics of an interest and $g$ indicates the goal. Given the statistics of a specific interest $e$, the function $f$ can map the trajectory into the corresponding goal.
>
> Our contribution is the same as in the introduction, including :
> 1. We draw attention to the goal-switching problem and argue that doing so is an essential step toward controllable policy.
> 2. We propose GMT that can solve this problem efficiently, whereas no other existing method could.
> 3. We propose an unsupervised clustering method that can efficiently perform trajectory clustering. Without clustering, each trajectory might be regarded as having different goals. However, we demonstrate that an unreasonable increase in clusters can harm performance (as shown in A.2.7) as there is no substantial difference between trajectories, for example, received 10 (as episode reward) and 10.1. Thus, we need a strategy to measure the similarity across different trajectories and group those with similar properties in terms of the statistics of interests.
>
> > My major concern about this paper is that the authors have not included any comparison of the proposed method to previously proposed baselines. Goal or return conditioned RL is a relatively mature subfield in RL with many prior works. Just to name a few, I believe the specific tasks solved in the paper can also be solved by prior methods such as Decision Transformers [1], Trajectory Transformers [2], HER [3], GCSL [4], RCP [5] and RvS [6]. Therefore, it is impossible to verify the empirical performance of the proposed method without comparing to some of these baselines.
>
> We move part of the results and explanations from the appendix to the section. We further put a table to summarize the performance of baselines (including RvS, Prompt-DT, and Decision Transformer) on our problem. We illustrate the final results in success rate at the beginning of Section 3. A few adjustments are made towards each baseline as their original architecture cannot directly be applied to our tasks (Section 3.1.2). In conclusion, none show comparable performance in goal-switching problems against GMT.
>
> > The description of the goal-switching problem can be improved since the problem of goal-switching is essentially the problem of achieving multiple goals in the term of goal-conditioned RL.
>
> The goal-switching can be seen as the problem of achieving multiple goals in one episode, which is not quite the same as the current multiple-goal-conditioned problem setting but is even more difficult.

---

### Decision · Program_Chairs · 2023-01-20

**Decision:**

Reject

**Justification For Why Not Higher Score:**

The paper has some interesting ideas but also significant weaknesses. The main weakness is that original paper had no comparisons to previously proposed methods (e.g. [1-6] in review S5ck and review 33J4); the revision includes some comparisons but on a limited set of 4 tasks all in simple mujoco locomotion tasks with low-dimensional observations. There are also some issues with clarity of the writing.

**Justification For Why Not Lower Score:**

n/a

**Metareview: Summary, Strengths And Weaknesses:**

The paper has some interesting ideas but also significant weaknesses. The main weakness is that original paper had no comparisons to previously proposed methods (e.g. [1-6] in review S5ck and review 33J4); the revision includes some comparisons but on a limited set of 4 tasks all in simple mujoco locomotion tasks with low-dimensional observations. There are also some issues with clarity of the writing.